# Are Civility Norms Morality Norms' Little Sister? The Truth Value That Lay Thinking Associates with Civility and Morality Social Norms

Armando Rodríguez-Pérez [1], Ramón Rodríguez-Torres [1], Verónica Betancor [1], Xing Jie Chen-Xia [1] and Laura Rodríguez-Gómez [2,*]

[1] Department of Cognitive, Social and Organizational Psychology, Faculty of Psychology, University of La Laguna, 38200 San Cristóbal de La Laguna, Spain
[2] Faculty of Health Sciences, Universidad Europea de Canarias, 38300 La Orotava, Spain
[*] Correspondence: laura.rodriguez@universidadeuropea.es

**Abstract:** Previous research shows that civility norms, such as morality norms, are necessary navigational charts to orient an individual in social life. However, there are no studies that establish the extent to which people consider civility norms as objective facts in a similar way to how many moral norms are considered. This research examines the perceived objectivity of civility norms in contrast to morality norms. The results show, firstly, that immoral norms are perceived to be significantly more objective than moral norms, but the opposite occurs with civility norms. Second, there is a high correlation between what participants consider objective and what they believe the rest of their community considers objective. However, this correlation was greater for immoral behaviors than for moral behaviors, and the opposite occurred for civility behaviors. Finally, participants estimated that the percentage of people in their group who engage in uncivil behaviors is higher than those who engage in immoral behaviors.

**Keywords:** civility; morality; social norms; moral objectivity; metaethics





Social life is regulated by normative systems that guide interactions and establish the necessary mechanisms to correct any transgressions that may occur. One of these normative systems is the moral rules—a set of standards that serve to distinguish between good and bad behaviors and prevent from harming others, the community, its symbols, or its institutions (Haidt 2001). However, in recent years, studies on the well-being of people in cities and organizations have led to a proliferation of studies on the norms of civility (e.g., Calhoun 2000; Cortina et al. 2017; Rodríguez-Gómez et al. 2021a, 2021b, 2022; Smith et al. 2010). Civility is considered "the lubricant that makes modern urban life possible" (Boyd 2006, p. 871). It is based on a firm commitment to the city and its inhabitants (Camps 2005). It upholds the importance of tolerance and concern for the feelings of others and seeks the protection of norms of mutual respect within social contexts (Calhoun 2000; Fritz 2017). Thus, civic norms include, on the one hand, manners and good citizenship, but they go further and also reflect the concern for the well-being of all community members and for the community in general (Forni 2002). Although transgressions of the civility rules are not considered properly immoral (e.g., not picking up the dog's excrement, not thanking someone who gives way, and being late for appointments), many researchers agree that they are a source of stress and a threat to well-being (Phillips and Smith 2003; Robin et al. 2007). Thus, do people perceive morality norms differently from civility norms? The present research aims to study how uncivil behaviors are perceived in comparison with immoral behaviors.

According to various authors (Goodwin and Darley 2008; Nichols 2004), one of the characteristics that differentiates morality norms is the degree of objectivity attributed

to them—that is, the degree to which people think that what is right and what is wrong is an objective truth in the same way that they think that 2 + 2 = 4. However, in the norms that regulate social life, there is no external criterion that allows for validating a statement; therefore, ethical disagreements are frequent. As a result, people might believe that a behavior is wrong, but that it is a personal opinion on which others might disagree. However, they could also believe that the wrongdoing is an objective truth and that anyone who disagrees is making a mistake. Therefore, could it be affirmed that morality norms are attributed more objectivity than civility norms, and that is why deviation from civility norms is more frequent than from morality norms?

As far as we know, no research has verified whether there are differences in the objectivity attributed to morality and civility norms. The study of the objectivity attributed to civility norms is important because this objectivity endows these norms with the motivational force that makes people feel better when they carry out behaviors that both they and others judge as objectively correct. In addition, they feel more legitimized to carry out actions of social control in the face of uncivil actions. Thus, the aim of this study is to fill this gap. To achieve this, we will research the effect of three factors that could highlight the differences and similarities between morality and civility norms.

First, the role of valence. It is possible that objectivity beliefs depend on the behaviors' positive or negative valence. In the field of moral standards, Goodwin and Darley (2012) found that beliefs about the wrongness or badness of negative moral actions (e.g., stealing, cheating) are considered more objective than beliefs about the rightness or goodness of positive moral actions (e.g., donating money to charities, contributing to environmental causes). Is it the same with the civility norms?

Second, the role of ingroup perceived consensus. In this sense, Goodwin and Darley's (2012) results confirmed the social anchoring of beliefs about moral objectivity. In fact, their results showed a high correlation between attributed objectivity and perceived consensus on morality behaviors. To what extent does this also apply to civility behaviors? Can it be said that the more people agree to consider a certain civility behavior bad or good, the greater the tendency to consider a civility belief to be objective?

Third, the role of moral identity. Various studies have found that people report that they engage in more "good" behaviors and fewer "bad" behaviors than "other people" do, and tend to emphasize their honesty more than their intelligence (Allison et al. 1989; Van Lange and Sedikides 1998). Does this occur in the same way in conducts of morality as in conducts of civility? Moreover, if an important ingredient of social identity is the moral value we give to the endogroup (Leidner and Castano 2012), is it possible that people tend to estimate that the ingroup will carry out more moral and civil than immoral and uncivil behaviors?

## 1. The Present Study: Overview and Hypotheses

According to the antecedents described above, this study aims to provide novel conclusions about civility norms and, specifically, the truth value attributed to these norms. These conclusions will help us understand why it is proving so difficult to get citizens to respect courtesy norms that improve the quality of life and well-being in cities. Specifically, the present research seeks to extend prior evidence on moral objectivity to beliefs about civility norms and to check whether civility norms follow the same metaethical pattern as morality norms. The hypotheses that guide this research are as follows: (1) It is expected that participants will attribute more objectivity to beliefs about morality behaviors than to beliefs about civility behaviors and more objectivity to beliefs about negative (immoral and uncivil) than positive (moral and civil) behaviors. (2) It is expected that objectivity about morality beliefs will be more supported by the ingroup consensus than civility behaviors and that negative behaviors will be more supported than positive ones. (3) It is expected that participants will estimate that a higher percentage of Spaniards perform positive than negative behaviors. Additionally, a significant positive correlation is expected between the objectivity attributed to morality and civility behaviors and the estimated percentage

of people in the ingroup who perform those behaviors. This correlation will be greater for morality behaviors than for civility behaviors. (4) It is expected that participants will estimate that they carry out more positive than negative behaviors in their daily lives. In addition, a significant positive correlation is expected between the estimated frequency of these behaviors and the objectivity attributed to morality and civility behaviors. In this case, it is very likely that the correlation will be higher for morality behaviors than for civility behaviors.

## 2. Method

### 2.1. Participants

No known previous research has examined the differences between civility and morality behaviors. G * Power (Faul et al. 2009) suggests that a sample of 125 participants enables detecting small effects of $f = 0.15$ assuming an alpha of 0.05 and a power of 0.95 for ANOVA (repeated measures and intra-subject interactions). A total of 126 Spanish university students (113 females) participated in this study, with ages ranging from 18 to 58 ($M = 18.89$; $SD = 3.70$), all of whom were residents in Spain. Students were offered college credit in Social Psychology.

### 2.2. Material and Procedure

After the participants' consent to participate in the study was obtained, they were informed that the aim of the questionnaire was to discover how people rate others' behavior. Specifically, they were told that they would be given different stories in which an anonymous person performed a behavior and they had to rate six questions:

*Objectivity attributed to the behavior.* To record the degree to which the participants felt that for each situation there was a morally correct way to respond, two questions were asked. The first question was, "To what extent is saying that this person has carried out a morally wrong (right) behavior an indisputable truth?" This was followed by a response scale from 1 (it is not an indisputable truth) to 10 (it is an indisputable truth). The second question was, "Now imagine a person who completely disagrees with your view on the matter; to what extent do you think that person is mistaken?" This question was followed by a scale from 0 (neither of us is mistaken) to 10 (the other person is clearly mistaken). In both cases, a higher score shows that there is only one morally correct answer and, therefore, indicates greater objectivity. Both measures correlated with each other—in the eight morality items the correlations ranged between $r(126) = 0.48$ and $r(126) = 0.76$, all $p < 0.01$; in the eight civility items the correlations ranged between $r(126) = 0.51$ and $r(126) = 0.79$, all $p < 0.01$; finally, in the two preferences (control) items the correlations were $r(126) = 0.46$ and 0.57 (consequently, these two measures were averaged to form a composite index of objectivity; Goodwin and Darley 2012).

*Opinion on the morality of the behavior.* The degree to which the participants agreed that the protagonist's behavior had been morally right or wrong was recorded. This question was included as a control measure that would allow for knowing if people distinguish when they agree with the goodness or badness of a behavior and when they believe that the goodness or badness of a behavior is an objective truth. Thus, a person might consider that "painting graffiti on street furniture" is wrong behavior, but believe that this is not an objective truth and that other people might disagree or that, in certain situations, graffiti is a form of artistic expression. To determine this, they were asked, "To what extent do you agree that HB has carried out a morally (in)correct behavior?" The question was followed by a 7-point scale (1: strongly agree; 7: strongly disagree).

*Perceived consensus.* This question asked the participants to estimate the percentage of Spaniards who would agree with the assessment made on the degree of objectivity of the behavior. Specifically, the participants were required to respond to the question, "What percentage of Spaniards would consider it an indisputable truth that doing what this person did is morally wrong?" This question was followed by a scale from 0 to 100, with the participant using a slider to answer.

*Percentage of Spaniards who would perform this behavior*. To determine the extent to which participants estimated that the behavior in the story was frequent among the members of their national group, the following question was asked: "In your view, what percentage of Spaniards would perform this behavior if they were in the same situation as X?" This question was followed by a sliding scale from 0 to 100.

*Frequency with which they had performed the behavior*. To determine the frequency with which the participants recalled having performed a similar behavior, the following direction was given: "Now indicate how often you have performed a behavior similar to that mentioned in the story". This question was followed by a scale from 0 (I have never done it) to 10 (I do it very frequently).

### 2.3. Vignettes

To select the behaviors, a preliminary study was carried out aimed at determining to what extent morality or civility norms were more relevant in evaluating a set of behaviors (see Supplementary Material S1). Based on the study, eight morality (four moral and four immoral) and eight civility (four civil and four uncivil) behaviors were selected. The four immoral behaviors included transgressions involving harm, unfairness, and willful dishonesty (e.g., "Online scam") while the four moral behaviors included altruistic and honest actions (e.g., "Helping an injured person on the road"). The four uncivil behaviors were actions that disrespected common property and good coexistence (e.g., "Not pick up the dog's droppings") while the four civil behaviors included actions of gratitude, consideration, and respect for others (e.g., "Disconnect the mobile at the cinema").

Together, the eight morality behaviors consistently represented situations sensitive to a judgment associated with morality standards ($\alpha = 0.82$), and the eight civility behaviors consistently represented situations sensitive to a judgment associated with civility standards ($\alpha = 0.79$).

In addition, as a control category, two stories were added that required preference judgments, where the subjective value of the responses predominated (Goodwin and Darley 2012). Specifically, the two stories alluded to preferences between two singers (Amy Winehouse vs. Lana del Rey) and the convenience (or inconvenience) of using liquid hydrogen in recipe preparation.

For each of the behaviors selected in the pilot study, and following Knutson et al.'s (2010) model, stories were drafted with a maximum of four sentences and a similar number of words in each. Overall, the stories, which averaged 59 words, did not differ significantly in the number of words, $F(1, 15) = 0.026$, $p = 0.876$, in the four story categories. This approach avoided confusion and ensured the stories would be easy to understand, making the answers to the questions following each story more reliable. In addition, a title headed each of the stories (e.g., "Disconnect the mobile at the cinema"). This was followed by the story itself (e.g., "On Saturday, CN met several friends to go to the movies to see an adventure film that recently won an Oscar. There were not many people in the theatre. Regardless, before sitting in their seat, CN disconnected their cell phone so as not to disturb anyone") (see Supplementary Material S2).

Vignettes were presented in a counterbalanced order, and the questionnaire was distributed among the university population using the Qualtrics online platform.

### 2.4. Data Analysis

To analyze the differences in the attribution of moral objectivity to behaviors related to morality (moral and immoral) and to civility (civil and uncivil), we computed the responses to 18 behaviors (4 moral, 4 immoral, 4 civil, 4 uncivil, and 2 preferences). Then, after we ensured that the data met the necessary assumptions for the analyses for each of these categories, we calculated the averages in "Objectivity attributed to the behavior", "Opinion on the morality of the behavior", "Perceived consensus", "Percentage of Spaniards who would perform this behavior", and "Frequency with which participants had performed the behavior".

With the scores on these variables, several repeated measures ANOVAs were carried out to contrast morality, civility, and preferences behaviors, as well as the meaning of the behaviors' valence in these relationships. These analyses were carried out to determine the attributed objectivity and the estimated support in the population. Moreover, several correlation analyses were carried out to specify the relationship between objectivity attributed to morality and civility behaviors with the percentage of people in the group who perform those behaviors, as well as with the frequency with which the participants perform the behaviors.

## 3. Results

### 3.1. Moral, Immoral, Civil, and Uncivil Behaviors and Attributed Objectivity

To learn if more objectivity is attributed to morality than to civility norms, we verified the degree of objectivity attributed to the conduct of morality, civility, and preferences. For this, a repeated measures ANOVA was carried out. The results show that beliefs about morality behaviors were perceived as being more objective ($M = 8.44$, $SD = 1.28$) than beliefs about civility behaviors ($M = 7.82$, $SD = 1.29$) and preferences, $M = 3.96$, $SD = 2.07$, $F(2, 125) = 403.5$, $p < 0.001$, $\eta^2_p = 0.763$. The contrast between pairs executed showed differences between the three types of behaviors taken two by two ($p < 0.001$).

In addition, to determine the impact of valence on morality and civility behavior, we performed a 2 (Type of behavior: morality vs. civility) × 2 (Valence: positive vs. negative) repeated measures ANOVA. The results showed a main effect of the type of behavior— $F(1, 125) = 81.59$, $p < 0.001$, $\eta^2_p = 0.395$—which was tempered by the interaction between behavior and valence, $F(2, 125) = 93.83$, $p < 0.001$, $\eta^2_p = 0.429$. Specifically, confirming Hypothesis 1, beliefs about immoral acts ($M = 8.68$, $SD = 1.34$) were perceived to be significantly more objective than beliefs about moral acts, $M = 8.19$, $SD = 1.37$, $t(125) = 6.19$, $p = 0.001$, $d = 0.551$, 95% CI [0.36, 0.74]. However, surprisingly, civility behaviors yielded inverse scores, because civil acts ($M = 8.14$, $SD = 1.38$) were perceived to be more objective than uncivil acts, $M = 7.50$, $SD = 1.45$, $t(125) = 6.07$, $p = 0.001$, $d = 0.541$, 95% CI [0.35, 0.73]. Furthermore, there were no differences in the objective value of positive moral behaviors and positive civil behaviors, $t(125) = 0.643$, $p = 0.521$. In contrast, more moral objectivity was attributed to the incorrectness of immoral behaviors than to that of uncivil behaviors, $t(125) = 12.77$, $p = 0.001$, $d = 1.138$, 95% CI [0.91, 1.36]. Table 1 displays the means in objectivity for each type of belief, including the result of the comparison of means between all of them.

**Table 1.** Means in objectivity and strength of agreement for each type of belief.

|  | MORALITY | | CIVILITY | |
|---|---|---|---|---|
|  | Positive | Negative | Positive | Negative |
| Objectivity | 8.19 a | 8.68 b | 8.14 a | 7.50 c |
| Strength of agreement | 6.71 a | 6.24 b | 6.79 c | 5.95 d |

*Note.* Cells that do not share a subscript are significantly different at $p < 0.001$.

Interestingly, the differences in the attribution of objectivity based on the valence of the behaviors do not arise from differences in overall strength of agreement. The 2 (Type of conduct: morality vs. civility) × 2 (Valence: positive vs. negative) ANOVA executed with the opinions ("To what extent do you agree that X has carried out a morally incorrect behavior?") resulted in a significant interaction—$F(1, 125) = 21.11$, $p < 0.001$, $\eta^2_p = 0.144$— which indicates that morality behaviors follow an inverse pattern. Thus, the degree of agreement with the statement about moral behaviors ($M = 6.71$, $SD = 0.49$) was higher than with that about immoral behaviors, $M = 6.24$, $SD = 1.32$, $t(125) = 4.18$, $p < 0.001$, $d = 0.372$, 95% CI [0.19, 0.55]. For civil behavior, the same response pattern is maintained in the attribution of objectivity and in the strength of the agreement. Specifically, the strength of the agreement is greater with civil behaviors ($M = 6.79$, $SD = 0.35$) than with uncivil behaviors, $M = 5.95$, $SD = 1.16$, $t(125) = 8.22$, $p < 0.001$, $d = 0.732$, 95% CI [0.53, 0.93]. Table 1

shows the mean scores relative to the degree of agreement for each type of belief, including the result of the comparison of means between all of them.

### 3.2. Perceived Consensus on Morality and Civility Behaviors and Attributed Objectivity

Another important aspect that allows us to identify the particularities of civility in relation to morality is the perceived consensus regarding the objectivity attributed to the morality, civility, and preference behaviors. We refer here to the estimated percentage of Spaniards who participants believed would agree with them on the objectivity attributed to the behaviors and, therefore, would share the view that there is only one correct assessment possible. The distribution of the estimated percentages is shown in Table 2. The repeated measures one-way ANOVA showed that the perceived consensus in relation to morality beliefs ($M = 77.30$, $SD = 12.58$) was higher than that of civility beliefs ($M = 74.87$, $SD = 13.55$) and preferences, $M = 43.05$, $SD = 19.80$, $F(2, 124) = 256.67$, $p < 0.001$, $\eta^2_p = 0.674$. The contrast between pairs showed differences between the three types of behaviors taken two by two ($p < 0.001$).

**Table 2.** Estimated percentage of Spaniards who would agree with the participant on the attribution of moral truth value for moral and civil behaviors and correlation with the attributed objectivity and agreement with personal opinion.

| | MORAL | | CIVIL | |
|---|---|---|---|---|
| | **Positive** | **Negative** | **Positive** | **Negative** |
| Perceived consensus (%) | 75.52 a | 79.16 b | 79.83 b | 70.17 c |
| Correlation between perceived consensus and objectivity | 0.488 ** | 0.504 ** | 0.582 ** | 0.403 ** |
| Correlation between perceived consensus and personal opinion on the correctness of the behavior | 0.284 ** | 0.070 | 0.386 ** | 0.076 |

*Note.* Cells that do not share a subscript are significantly different at $p < 0.001$. ** $p < 0.01$.

Furthermore, to test whether objectivity about morality norms is more supported than civility norms by the ingroup consensus, and negative behaviors are more supported than positive ones, we conducted a 2 (Type of conduct: morality vs. civility) × 2 (Valence: positive vs. negative) repeated measures ANOVA. The analysis showed two significant main effects, $F(1, 125) = 12.49$, $p = 0.001$, $\eta^2_p = 0.091$ for behaviors and $F(1, 125) = 17.88$, $p < 0.001$, $\eta^2_p = 0.125$ for valence. However, contrary to what was expected in Hypothesis 2, the significant interaction—$F(1, 125) = 103.43$, $p < 0.001$, $\eta^2_p = 0.453$—showed, on the one hand, that the participants estimated that there would be more Spaniards who agreed with them regarding the truth value of immoral behaviors ($M = 79.16$, $SD = 12.05$) than of moral behaviors, $M = 75.52$, $SD = 15.03$, $t(125) = 3.83$, $p < 0.001$, $d = 0.341$, 95% CI [0.16, 0.52]. On the other hand, we expected that civility behaviors would follow the reverse pattern. Participants estimated the percentage of Spaniards who would agree with them regarding the truth value of uncivil behaviors ($M = 70.17$, $SD = 14.36$) as significantly lower than for civil behaviors, $M = 79.83$, $SD = 14.90$, $t(125) = 9.81$, $p < 0.001$, $d = 0.874$, 95% CI [0.67, 1.08]. Consequently, the participants predicted that there would be more consensus in qualifying as morally wrong an immoral action than an uncivil action—$t(125) = 9.67$, $p < 0.001$, $d = 0.862$, 95% CI [0.66, 1.06]—but more consensus in qualifying as morally right a civil action than a moral action, $t(125) = 4.63$, $p < 0.001$, $d = 0.413$, 95% CI [0.23, 0.59].

Interestingly, this perceived consensus regarding objectivity correlates significantly with the attribution of objectivity for both immoral—$r(126) = 0.504$, $p < 0.001$—and moral behaviors—$r(126) = 0.488$, $p < 0.001$—and both uncivil—$r(126) = 0.403$, $p < 0.001$—and civil behaviors—$r(126) = 0.582$, $p < 0.001$. However, it is striking that this result is different with respect to the strength of agreement about the (in)correctness of the behaviors. Specifically, whereas a significant correlation was found between personal opinion about the correctness of the behaviors and the percentage of Spaniards who would attribute objectivity ($r = 0.284$,

$p < 0.01$ for moral and $r = 0.386$, $p < 0.01$ for civil behaviors), no correlation was found with respect to immoral ($r = 0.07$) and uncivil behaviors ($r = 0.08$). That is, participants believe that what they consider good coincides with what other Spaniards would consider to be objectively good. On the other hand, there is no relationship between what participants think is bad and what they believe the Spanish think is objectively bad. To explore this, a regression analysis was carried out with personal opinion and perceived consensus as predictive variables, and moral objectivity as a criterion variable for moral behaviors. This analysis showed that the two variables explained 25.5% of the variance, $F(2, 125) = 22.39$, $p < 0.001$. However, the perceived consensus was the only variable that contributed a significant value ($B = 0.049$, $\beta = 0.483$, $p < 0.001$). This same regression analysis relative to civil behaviors explained a similar percentage of the variance of objectivity judgments (25.6%), $F(1, 125) = 22.46$, $p < 0.001$. However, comparing the standardized regression coefficient (Beta), we found that the consensus variable ($B = 0.045$, $\beta = 0.469$, $p < 0.001$) contributed more significantly to the objectivity variance than did personal opinion ($B = 0.350$, $\beta = 0.172$, $p = 0.029$, $z = 2.63$, $p \leq 0.01$).

### 3.3. Perceived Percentage of the Ingroup That Would Engage in Morality and Civility Behaviors and Attributed Objectivity

What percentage of Spaniards do participants consider would behave as described in the stories if they were in a similar situation? Does this estimate have something to do with the assessment made about the statements' objectivity? It is quite possible that there is a strong tendency to protect the moral status of the ingroup and, consequently, their responses are biased in favor of moral and civil behaviors. To confirm this, we performed a 2 (Type of conduct: morality vs. civility) $\times$ 2 (Valence: positive vs. negative) repeated measures ANOVA. The analysis showed the significant effects of the two study variables. First, and according to Hypothesis 3, there was a main effect of valence. Participants estimated that Spaniards perform more positive behaviors ($M = 60.37$, $SD = 13.21$) than negative ones, $M = 48.59$, $SD = 14.08$, $F(1, 125) = 49.01$, $p < 0.001$, $\eta^2_p = 0.282$. Furthermore, the type of behavior was also significant. Participants estimated that their national ingroup performs more civility ($M = 61.01$, $SD = 10.70$) than morality behaviors, $M = 47.95$, $SD = 11.38$, $F(1, 125) = 213.82$, $p < 0.001$, $\eta2_p = 0.631$. The interaction was not significant. Consequently, the responses showed that the participants estimated civil behaviors ($M = 67.51$, $SD = 14.20$) to be more frequent than uncivil behaviors—$M = 54.52$, $SD = 15.15$, $t(125) = 7.26$, $p < 0.001$, $d = 0.647$, 95% CI [0.45, 0.84]—and moral behaviors ($M = 53.23$, $SD = 15.08$) to be more frequent than immoral behaviors, $M = 42.66$, $SD = 15.08$, $t(125) = 5.22$, $p < 0.001$, $d = 0.465$, 95% CI [0.28, 0.65].

Regarding the relationship between participants' estimates of Spaniards who would perform these behaviors and their attributed objectivity, the analysis showed a significantly higher correlation for civil statements—$r(126) = 0.352$, $p < 0.001$—than for moral statements, $r(126) = 0.200$, $p = 0.025$, $z = 2.064$, $p = 0.019$. That is, the greater the percentage of people estimated to engage in the behavior, the greater the objectivity attributed to that behavior. However, for both immoral behaviors—$r(126) = -0.162$, $p = 0.070$—and uncivil behaviors—$r(126) = -0.170$, $p = 0.057$—the relationship was only marginally significant (see Table 3).

**Table 3.** Frequency of own and others' behavior and correlation with attributed objectivity.

| | MORAL | | CIVIL | |
|---|---|---|---|---|
| | **Positive** | **Negative** | **Positive** | **Negative** |
| Percentage of others' behavior (%) | 53.23 b | 42.66 c | 67.51 a | 54.52 b |
| Frequency of own behavior | 4.40 b | 0.61 d | 7.88 a | 2.10 c |
| Correlation others' percentage/attributed objectivity | 0.200 * | −0.162 | 0.352 ** | −0.170 |
| Correlation own frequency/attributed objectivity | 0.140 | −0.341 ** | 0.442 ** | −0.336 ** |

*Note.* Cells that do not share a subscript are significantly different at $p < 0.001$. * $p < 0.05$. ** $p < 0.01$.

*3.4. Estimated Frequency of Self-Performed Morality and Civility Behaviors and Attributed Objectivity*

The strong tendency toward self-aggrandizement suggests that asking participants how often they would perform the behaviors referred to in this research produces a bias in favor of positive behaviors and against negative behaviors. To analyze behaviors according to their valence, we conducted a 2 (Type of conduct: morality vs. civility) × 2 (Valence: positive vs. negative) repeated measures ANOVA. As expected in Hypothesis 4, the analysis showed the main effects of the two variables—$F(1, 125) = 523.63$, $p < 0.001$, $\eta^2_p = 0.807$ for behaviors and $F(1, 125) = 753.25$, $p < 0.001$, $\eta^2_p = 0.858$ for valence—as well as a significant interaction, $F(1, 125) = 68.90$, $p < 0.001$, $\eta^2_p = 0.355$.

The contrast analysis showed that on the scale (0: I have never done it; 10: I do it very frequently), the participants gave a lower score to immoral behaviors ($M = 0.61$, $SD = 1.13$) than to moral behaviors, $M = 4.40$, $SD = 2.35$, $t(125) = 17.78$, $p < 0.001$, $d = 1.58$, 95% CI [1.32, 1.85] (see Table 3). The same was true for civility behaviors. The frequency of uncivil behaviors ($M = 2.09$, $SD = 1.72$) was significantly lower than that of civil behaviors, $M = 7.89$, $SD = 1.48$, $t(125) = 27.40$, $p < 0.001$, $d = 2.44$, 95% CI [2.09, 2.79]. Furthermore, the analysis of contrasts between morality and civility behaviors showed that the participants performed more civil than moral behaviors—$t(125) = 17.97$, $p < 0.001$, $d = 1.60$, 95% CI [1.34, 1.86]—and more uncivil than immoral behaviors, $t(125) = 12.12$, $p < 0.001$, $d = 1.08$, 95% CI [0.86, 1.30].

It is more interesting to verify the correlation between frequency and objectivity. The results showed a significant association for civility behaviors, $r(126) = 0.442$, $p < 0.001$ for civil behaviors and $r(126) = -0.336$, $p < 0.001$, and for uncivil behaviors. In other words, the more participants acknowledged that they engage in civility behaviors, the more they tended to consider that they are objectively correct behaviors. The opposite also occurred. The less they engaged in uncivil behavior, the more morally wrong they considered it to be. This same relationship was found with immoral behaviors—$r(126) = -0.341$, $p < 0.001$—but not with moral behaviors, $r(126) = 0.140$, $p = 0.117$ (see Table 3).

## 4. Discussion

The results of the present study provide a diffused picture of the boundaries separating civility norms from morality norms. Several conclusions can be drawn from the analysis of the answers. First, morality norms are perceived as more objective than civility norms. Furthermore, in line with Goodwin and Darley's (2008, 2012) findings, immoral norms are perceived to be significantly more objective than moral norms (Hypothesis 1). However, notably, the opposite occurs with civility norms. Here, civil norms are considered more objective than uncivil norms.

The existence of different patterns in the objectivity attributed to morality and civility statements could be supported by the two systems of moral regulation proposed by Janoff-Bulman et al. (2009), specifically in the differentiation between prescriptive and proscriptive norms. According to these authors, these two types of norms reflect the approximation-avoidance motivational distinction, where proscriptive morality is sensitive to negative results and, therefore, based on inhibitory processes that seek to avoid punishment, whereas prescriptive morality is sensitive to positive results and based on activation processes that seek to obtain rewards.

In this sense, we can suppose, on the one hand, that in morality behaviors, the proscriptive norm will prevail over the prescriptive one; that is, what should not be done will prevail over what should be done, affecting immoral behaviors more than moral behaviors (Anderson et al. 2020). For example, the behavior "deceiving someone" would be perceived as objectively immoral (Stanley et al. 2020; Wright and Pölzler 2021) and deserving of strong social and legal sanctions, whereas the behavior "being honest with someone" would not have the same objective force as a moral action. On the other hand, for civility, the prescriptive norm will prevail over the proscriptive norm; that is, what should be done will prevail over what should not be done, affecting civil behaviors more than uncivil ones. Therefore, a person is more likely to classify the behavior "giving your seat on

the bus to a pregnant woman" as objectively good as opposed to considering "remaining seated" objectively bad.

The modern conception of civility could also explain the differences in objectivity found between civic and uncivil behaviors. The study of civility from a political perspective in relation to the communicative practices of civil discourse has made it possible to broaden the limits previously established in relation to the meaning of what is civic and what is not. Generally, civility has been associated with respect for the rules of coexistence, initially in relation to courtesy and later, from a broader perspective, with the relationships that exist in the community between people, and between them and the community's own resources. Therefore, from this vision of civility, any uncivil behavior, that is, that goes against social concord, would be valued as negative, and would facilitate an attitude of social disapproval on the part of the community (Bejan 2017). However, new theoretical lines have pointed out the positive character of incivility. Civility implies a commitment to the community (Camps 2005). From this conception, civility can be understood as those behaviors that, based on respect, allow democratic relations among its members (the classical conception of civility) and would also include those behaviors that imply an improvement for the advancement of the community. From this last perspective, some uncivil behaviors could fit into that definition. In this sense, there are uncivil behaviors that would be supported by the primary objective of breaking with norms or situations that harm members of society or, more generally, the advancement of society itself (Edyvane 2020). This positive conceptualization of incivility arises in response to the idea that civility throughout history has functioned as a form of repression against minority groups by allowing the hegemony of the sovereign group (Applebaum 2021; Keith and Danisch 2020; Zamalin 2021). This ambiguity when it comes to defining uncivil behavior depending on whether it is really negative (because it affects the links between people in the community by not respecting its members) or, on the contrary, it is positive (because it seeks to improve community life), could determine that uncivil behaviors are not as objective as civic ones.

Second, this study establishes that the objectivity attributed to morality and civility statements is not alien to the consensus expected in the ingroup. The analysis of the responses showed a high correlation between what participants consider objective and what they believe the rest of their community considers objective. Furthermore, in line with Hypothesis 2, this correlation was greater for immoral behaviors than for moral behaviors, and the opposite occurred for civility behaviors. This result confirms the role of social anchoring that supports norms that are considered social norms about what is morally correct or incorrect (Ellemers et al. 2019). Ellemers et al. (2019) asserted that moral convictions are imperative mandates of the community, showing what everyone "ought" to or "should" do. In a similar vein, Haidt (2001) argued that moral intuitions derive from implicit learning of peer group norms and cultural socialization, and, consequently, shared ideas about right and wrong vary, depending on the individual, cultural, religious, or political context in which they are defined (Giner-Sorolla 2012; Haidt and Graham 2007; Haidt and Kesebir 2010; Rai and Fiske 2011; Wright 2021). Even more interesting is that the participants in our study estimated that more than 70% of their community would support both their morality and civility beliefs, a clear example of the effect of false consensus on beliefs about what is right and what is wrong (Ross et al. 1977).

Interestingly, personal opinion and perceived social consensus correlate only when the "correctness" of positive behaviors is considered, not when the "incorrectness" of negative behaviors is assessed. That is, participants consider that there is a high correspondence between what they consider correct (personal opinion) and what others consider correct (perceived consensus), but not between what they consider incorrect and what others consider incorrect. Consequently, they infer that the majority of society agrees with them that "giving up your seat to an elderly person", for example, is correct behavior. However, for negative behaviors such as "deceiving someone", they do not perceive the same correspondence between their personal opinion and the majority view.

We might ask, why does the false consensus bias occur more in beliefs about moral and civil behaviors than about immoral and uncivil behaviors? One possible explanation has to do with the greater flexibility associated with the ethical assessment of negative behaviors. In other words, this is basically due to, on the one hand, the high prevalence of "ordinary" unethical behaviors performed by people who value morality but act immorally when they have the opportunity to do so (Ayal and Gino 2011). On the other hand, it is due to the need to resolve the "ethical dissonance" that follows this action to preserve a moral self-image (Barkan et al. 2010). One way to mitigate this dissonance has to do with social comparisons. Thus, by observing the unethical behavior of other people, individuals learn the social norms related to dishonesty (Campbell 1964). This learning compels people to incorporate contextual factors (e.g., "If others are doing it, I can do it too") when they behave unethically, thus, reducing the ethical dissonance they experience by making the negative behavior less bad. Consequently, negative behaviors (e.g., "throwing papers into the street" or "deceiving others on the Internet") are subject to a more flexible ethical assessment and have a more variable ethical status in the population.

Third, another factor that contributes to differentiating between morality and civility statements has to do with the frequency with which behaviors are carried out by the individual and the percentage of Spaniards who would perform such behaviors (Hypotheses 3 and 4). As a whole, participants considered that they perform more civility than morality behaviors, and they believed the same about the behaviors of others. Furthermore, they considered that they hardly engage in negative behaviors (neither uncivil nor immoral); that is, they tend to attribute virtuous behaviors to themselves, thus, protecting their moral status and drawing of themselves a profile of someone who does not tolerate fraudulent actions that damage their moral identity (Stavrova et al. 2013).

Although our study does not make explicit comparisons between oneself and others, our results are aligned with those of Messick et al. (1985), according to whom people tend to safeguard their moral identity in relation to others, amplifying the difference in the type of unfair behaviors performed. Specifically, people tend to associate low-intensity unethical behaviors with themselves, while attributing a greater number of unethical acts to others, including immoral acts such as cheating, stealing, destroying, shoplifting, etc. (Messick et al. 1985). According to these authors, these results suggest that the self-serving bias is better described by the belief that "I am less evil than you" than by "I am holier than you" (Allison et al. 1989; Epley and Dunning 2000).

In a similar vein, participants estimated that the percentage of people in their group who engage in moral and civil behaviors is higher than those who engage in immoral and uncivil behaviors. However, they considered that more than half of the population would carry out uncivil behaviors, whereas less than half of the population would carry out immoral behaviors if they were in a situation similar to the person in the vignette. This result qualifies what was anticipated in Hypothesis 3, which did not establish differences between beliefs regarding positive and negative behaviors. However, this asymmetry is interesting because it could be motivated by the desire to safeguard the morality of the ingroup. Thus, in a context where the ingroup transgresses social norms, are we more willing to say that the ingroup is immoral or uncivil? For our participants, the second is the dominant response. According to Leidner and Castano (2012), to protect their ingroups, people can minimize the immorality of behaviors by changing the moral framework. In this sense, people trust that changing an infraction of the moral foundation of harm to one of disobedience to authority reduces the bad behavior (Haidt 2001). According to the authors, this "morality shifting" need not be considered a deliberate and thoughtful strategy, but rather a form of self-serving.

Furthermore, the correlation analyses between attributed objectivity and personal frequency, on the one hand, and between attributed objectivity and percentage of the population that would carry out this behavior, on the other, yielded different results. The more objective the participants consider beliefs about the immoral and uncivil behaviors to be, the less frequently they perform them. The opposite occurs with civil behaviors.

The more objectively correct participants consider them to be, the more frequently they perform them. This relationship was not observed with moral behaviors. In addition, the relationship only occurs in positive (moral and civil) behaviors, so that the more people perform the behavior, the more objectively correct it is perceived to be. This is not observed with negative (immoral and uncivil) behaviors.

Taken together, the results of this study show that, in contrast to morality, for lay thinking, it is civil behaviors and not uncivil behaviors that are attributed more truth value. That is, lay thinking is more sure that giving the seat to an older person is right while doubting the incorrectness of not giving the seat to an older person. This uncertainty regarding the objectivity of what is uncivil is manifested in that it is the uncivil behavior for which less social consensus is perceived. It should not be surprising, therefore, that many uncivil behaviors do not receive open disapproval from citizens, and that there is little social consensus regarding their moral wrongdoing. Consequently, if in the realm of civility it is the correction of civil behaviors that has greater objectivity, it is logical that many people consider such behaviors a greater source of personal virtue than moral behaviors. In fact, the participants considered that both they and their ingroup engaged in more civil behaviors than moral behaviors and less immoral than uncivil behaviors. Furthermore, this behavioral pattern is associated with the attribution of objectivity to these behaviors.

*Limitations and Recommendations for Future Research*

The present study has several limitations that must be pointed out. First, our research explores participants' reactions to a limited number of moral and civil behaviors. The results found might not be generalizable to all moral and civil behaviors.

Second, the present research was carried out with Spanish university students. Although the results coincide with the findings of other studies conducted in the Western cultural context (Clifford et al. 2015; Goodwin and Darley 2012), it is possible that studies with samples of non-university participants or participants from other cultural contexts would give rise to different results.

Third, in relation to the choice of behaviors, the participants were asked about norms that were already considered moral or immoral beforehand, that is, about norms for which they have a previously defined position. In this sense, there may be a potential problem of endogeneity. New studies could explore alternative procedures in the selection of behaviors, such as asking about societies with novel norms (e.g., Dahl and Waltzer 2020) and, thus, measuring the morality or civility of the norm that is related to objectivity.

Finally, our study lacks a functional orientation referring to the value of the current naïve division between morality and civility and the role that culture and respect for the community play as moral norms (Rodríguez-Pérez 2020). This involves exploring whether attributing objectivity to moral beliefs increases the potential for conflict in intergroup relationships. In this sense, the differences between communities and social groups can become more intractable to the extent that each side defends an objective view of the truth of its own beliefs.

Future studies should investigate the extent to which the judgments of what is morally good versus bad depend on culturally defined virtues, that is, to what extent the shared ideas about good and evil vary depending on the cultural, religious, or political context in which they are defined (Giner-Sorolla 2012; Haidt and Graham 2007; Haidt and Kesebir 2010; Rai and Fiske 2011). This is interesting because it is possible that what in our cultural space is considered an incivility, such as "refusing to wear a school uniform", could be considered in certain cultures a moral violation, on the same level as "deception or cheating" (Buchtel et al. 2015; Haidt et al. 1993). Furthermore, it would be helpful to incorporate political orientation and support for moral foundations in these studies because the results of other research show that people with different ideological views will consider a behavior a moral violation on the basis of the moral foundation that it transgresses. It is, therefore, a question of determining whether accusations of immorality against those who transgress norms of civility differ depending on one's political orientation and moral beliefs.

**Supplementary Materials:** The following supporting information can be downloaded at: https://www.mdpi.com/article/10.3390/socsci11120568/s1, Supplementary Materials S1: Pilot study for the selection of behaviors, Supplementary Materials S2: Vignettes for each type of behaviors selected in the pilot study. (Chadwick et al. 2006) cited in the Supplementary Materials.

**Author Contributions:** Conceptualization, A.R.-P., R.R.-T. and L.R.-G., data curation, A.R.-P. and R.R.-T.; formal analysis, A.R.-P. and R.R.-T.; funding acquisition, A.R.-P. and V.B.; investigation, A.R.-P.; methodology, A.R.-P. and R.R.-T.; project administration, A.R.-P. and V.B.; resources, A.R.-P. and V.B.; software, L.R.-G., X.J.C.-X., A.R.-P. and R.R.-T.; supervision, A.R.-P. and L.R.-G.; validation, L.R.-G.; visualization, L.R.-G. and X.J.C.-X.; writing—original draft, A.R.-P., R.R.-T., V.B. and L.R.-G.; writing—review and editing, L.R.-G. and X.J.C.-X. All authors have read and agreed to the published version of the manuscript.

**Funding:** This research was supported by the Spanish Ministry of Science and Innovation under grant PID2019-108217GB-100.

**Institutional Review Board Statement:** The relevant ethical guidelines were followed. The study was conducted in accordance with the ethical standards of the Ethics Committee of the University of La Laguna for studies involving human participants, and approved by the Ethics Committee on Research and Animal Welfare of the University of La Laguna (CEIBA2021-0458).

**Informed Consent Statement:** Informed consent was obtained from all subjects involved in the study.

**Data Availability Statement:** The datasets presented in this study are available for download at https://osf.io/myswt/?view_only=bbaa682f1a814f50acd7224f64dbe4a0 (accessed on 13 September 2022).

**Conflicts of Interest:** The authors declare no conflict of interest. The funders had no role in the design of the study; in the collection, analyses, or interpretation of data; in the writing of the manuscript, or in the decision to publish the results.

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
