# Peer review of "Are Civility Norms Morality Norms’ Little Sister? The Truth Value That Lay Thinking Associates with Civility and Morality Social Norms"

_socsci, doi:10.3390/socsci11120568_

Round 1

Reviewer 1 Report

This is a well-done study looking at the objectivity of moral and civil norms. Both were basically found to be judged as objective, but moral norms are slightly less objective than immoral norms and vice versa with civil norms. Because of the way moral and civil norms are chosen, there is a potential endogeneity problem given that the subjects are being asked about norms that they already take to be moral or immoral, rather than those that they might not have a stance on or not. A better approach might be to follow Dahl & Waltzer 2020 and ask questions about people in other societies, perhaps with novel norms to gauge how much it is the "moralness" or "civilness" of the norm that is related to objectivity, rather than how much the students just agree with the claims themselves. That said, the evidence here does seem to support the basic claims being made, but it would be hard to know how to generalize from them or how to make general claims about moral/civil norms and objectivity. 

Author Response

We appreciate this comment. We find it very interesting to take this perspective into account for future studies in which we have to compare norms. We have decided to include this perspective in the manuscript as a limitation and proposal for future study.

Reviewer 2 Report

I think this essay is very well done and cogently argued. I also think it intervenes in a crucial current conversation. I'm happy to see the author(s) taking up questions about civility, as they seem especially important in our moment. I think the method was very clear, the presentation of results was very clear. My only issue is with the framing of the essay. I think the author(s) portray civility in rather simplistic terms, and this is a problem that needs to be addressed before publication. Here's what I mean: I don't think civility amounts to simply good manners. Some recent scholarship on civility offers a more complicated story (Stephen Carter's book on civility, Alex Zamalin's Against Civility, Danisch & Keith's Beyond Civility, and Bejan's Mere Civility book). So the author(s) I think need a more careful explication of what they mean by civility and a more careful positioning of their conception of civility within this more robust conversation about what constitutes civility. I feel like this would also help qualify the results and discussion sections of this paper. Perhaps some other ways to think through the meaning of the data would emerge if there was a more complex sense of what civility means. 

Author Response

Thank you very much for your suggestion, we completely agree. We have added a paragraph in the discussion on how the modern conceptualization of civility may be explaining the differences obtained between civic behavior and uncivil behavior. This comment has improved the quality of the manuscript discussion, thank you.
